# Genomic Landscape of Branchio-Oto-Renal Syndrome through Whole-Genome Sequencing: A Single Rare Disease Center Experience in South Korea

**DOI:** 10.3390/ijms25158149

**Published:** 2024-07-26

**Authors:** Sung Ho Cho, Sung Ho Jeong, Won Hoon Choi, Sang-Yeon Lee

**Affiliations:** 1Department of Otorhinolaryngology-Head and Neck Surgery, Seoul National University Hospital, Seoul National University College of Medicine, Seoul 03080, Republic of Korea; sam809@snu.ac.kr (S.H.C.); dennysh@naver.com (S.H.J.); wonhchoi@kakao.com (W.H.C.); 2Department of Genomic Medicine, Seoul National University Hospital, Seoul National University College of Medicine, Seoul 03080, Republic of Korea; 3Sensory Organ Research Institute, Seoul National University Medical Research Center, Seoul 03080, Republic of Korea

**Keywords:** BOR/BO syndrome, whole-genome sequencing (WGS), molecular diagnosis, structural variations (SVs)

## Abstract

Branchio-oto-renal (BOR) and branchio-otic (BO) syndromes are characterized by anomalies affecting the ears, often accompanied by hearing loss, as well as abnormalities in the branchial arches and renal system. These syndromes exhibit a broad spectrum of phenotypes and a complex genomic landscape, with significant contributions from the *EYA1* gene and the *SIX* gene family, including *SIX1* and *SIX5*. Due to their diverse phenotypic presentations, which can overlap with other genetic syndromes, molecular genetic confirmation is essential. As sequencing technologies advance, whole-genome sequencing (WGS) is increasingly used in rare disease diagnostics. We explored the genomic landscape of 23 unrelated Korean families with typical or atypical BOR/BO syndrome using a stepwise approach: targeted panel sequencing and exome sequencing (Step 1), multiplex ligation-dependent probe amplification (MLPA) with copy number variation screening (Step 2), and WGS (Step 3). Integrating WGS into our diagnostic pipeline detected structure variations, including cryptic inversion and complex genomic rearrangement, eventually enhancing the diagnostic yield to 91%. Our findings expand the genomic architecture of BOR/BO syndrome and highlight the need for WGS to address the genetic diagnosis of clinically heterogeneous rare diseases.

## 1. Introduction

The branchio-oto-renal (BOR) and branchio-otic (BO) syndromes represent a complex spectrum of rare, genetically heterogeneous conditions characterized by anomalies affecting the ears, branchial arches, and renal system [1,2] Certain individuals may display symptoms typical of BOR/BO syndrome but lack renal abnormalities, leading to diagnoses of branchio-oto syndrome-1 [3] (BOS1; OMIM#602588) or branchio-oto syndrome-3 [4] (BOS3; OMIM#608389). Diagnostic criteria for these syndromes encompass both major and minor clinical features [5]: Major criteria include deafness (noted in 98.5% of cases), branchial anomalies (49–73%), preauricular pits (53–83%), and renal anomalies (38–70%), while minor criteria consist of anomalies in the external, middle, and inner ears, as well as preauricular tags. Moreover, some individuals exhibit atypical presentations that do not conform entirely to the standard diagnostic criteria despite harboring pathogenic variants in genes linked to the BOR/BO syndromes. These syndromes demonstrate a high penetrance for hearing impairment, affecting over 90% of individuals [5,6], with hearing loss presenting as mixed (50%), conductive (30%), or sensorineural (20%), ranging from mild to profound in severity.

Manifesting in roughly 1 in 40,000 individuals within European demographics and responsible for 2% of profound childhood deafness [7], this syndrome demonstrates autosomal dominant inheritance with notable variability in expression and high genetic penetrance [7,8,9]. The genetic landscape of BOR/BO syndrome is complex, with major contributions from the *EYA1* gene [6,10,11] and the *SIX* gene family, including *SIX1* [12] and *SIX5* [13]. It is well known that EYA1 binds to SIX1 and SIX5 to form a bipartite transcription factor [14]. Especially, the SIX1 protein binds to the Eya domain of EYA1 using its Six domain and concurrently binds to DNA elements with its DNA binding homeodomain to form the EYA1-SIX1-DNA complex [15]. In turn, the EYA1-SIX1-DNA complex regulates organogenesis, including the branchial arch, otic, and renal systems [10,16]. *EYA1* mutations are the most prevalent [10,11], affecting 40–75% of cases, whereas *SIX1* mutations are found in a smaller portion of cases (3.0–4.5%) [12,16]. Variants in *SIX5* also play a role [13,17], though they account for only 0–3.1% of incidences.

Since the 2010s, the advent of high-throughput next-generation sequencing (NGS) technologies has significantly advanced our understanding of the genetic underpinnings of diseases [18,19]. Despite the genetic and phenotypic complexity of the syndrome, which complicates the accuracy of diagnoses, NGS has emerged as the preferred diagnostic tool due to its capacity to screen a wide range of genetic loci. Targeted panel sequencing (TPS) and whole-exome sequencing (WES) have been routinely used with varying success rates [20,21,22]. However, recent advancements in sequencing technologies have made whole-genome sequencing (WGS) more accessible [23], offering a broader and more detailed analysis of genomic variants than targeted approaches [24]. These developments have greatly expanded the potential for exploring the intricate genetic landscape of BOR/BO syndrome.

In this study, we explored the genomic landscape of 23 unrelated Korean families with typical or atypical BOR/BO syndrome using a stepwise approach. This stepwise approach included targeted panel sequencing and exome sequencing (Step 1), exome sequencing-based copy number variations (CNV) screening coupled with multiplex ligation-dependent probe amplification (MLPA) (Step 2), and WGS (Step 3). This comprehensive analytical framework, aided by WGS, has enhanced the molecular diagnostic yield of BOR/BO syndrome compared to diagnostic rates from previous studies. Consequently, this expands our understanding of the genomic architecture of BOR/BO syndrome and highlights the need for WGS implementation to address the genetically undiagnosed clinically heterogeneous rare diseases, including those with BOR/BO syndrome.

## 2. Results

### 2.1. Cohort Description and Clinical Phenotypes

In our cohort study, a structured genetic testing protocol was applied to 41 patients with clinical suspicion of BOR/BO syndrome from 23 families to identify segregation with pathogenic variants linked to *EYA1*, *SIX1*, and *ANKRD11* genes (Table 1). When examining the phenotypes that belong to the major criteria of BOR/BO syndrome, firstly, 40 (98%) patients experienced hearing loss—including conductive, mixed, and sensorineural. Branchial anomalies, preauricular pits, and renal anomalies were observed in twenty-seven (66%), thirty-four (83%), and six (15%) patients, respectively. Additional phenotypes corresponding to the minor criteria of BOR/BO syndrome, including external auditory canal (EAC; eight patients, 20%), middle ear (22 patients, 54%), and inner ear anomalies (sixteen patients, 39%), were observed in a smaller subset of the group. Patients who meet three of the major criteria (hearing loss, preauricular pits, branchial anomalies, renal anomalies, or auricular deformities), two of the major criteria with two minor criteria (inner ear anomalies, middle ear anomalies, external auditory canal anomalies, preauricular tags, facial asymmetry, or palatal anomalies), or one of major criterion with an affected first-degree relative meeting the criteria for BOR/BO syndrome can be clinically diagnosed with typical BOR/BO syndrome [10,25]. However, there are some patients who do not satisfy the clinical criteria for typical BOR/BO syndrome despite carrying a pathogenic or likely pathogenic variant of BOR/BO syndrome. Such patients are diagnosed with atypical BOR/BO syndrome [17,26,27]. According to the previously mentioned clinical diagnostic criteria of BOR/BO syndrome, 63% and 34% of the patients were diagnosed with typical and atypical BOR/BO syndrome in this study, respectively (Table 1).

### 2.2. Stepwise Molecular Diagnostic Yield

The stepwise genetic diagnostic yield is shown in Figure 1. Initially, targeted panel sequencing and exome sequencing were utilized to identify causative variants across the BOR/BO cohort. A total of 18 of the 23 families were genetically diagnosed with exome sequencing (molecular diagnostic yield = 78.3%), and their variants included single-nucleotide variants (SNVs), such as missense (seven families), nonsense (four families), and splicing variants (one family), or indel mutations (six families). For patients who remained undiagnosed after targeted panel sequencing and exome sequencing (Step 1), exome sequencing-based CNV screening coupled with MLPA (Step 2) was applied. One subject was further genetically diagnosed in Step 2 (cumulative molecular diagnostic yield = 82.6%). Finally, WGS was applied to three patients who remained genetically undiagnosed after Step 2, and two of these subjects were further genetically diagnosed in Step 3 (cumulative molecular diagnostic yield = 91.3%). WGS improved the molecular diagnostic yield by 8.7%. The molecular diagnostic yield of the stepwise genomic pipeline is shown in Figure 1. Overall, genetic diagnosis was made in 21 (91.3%) of the 23 unrelated families with clinical suspicion of BOR/BO syndrome. All variants described herein were classified as pathogenic or likely pathogenic according to ACMG-AMP guidelines [28]. The molecular diagnostic yield of typical BOR/BO syndrome was 96%, whereas the molecular diagnostic yield of atypical BOR/BO syndrome was 93%. As shown in Table 1, most of the atypical BOR/BO syndrome patients had *SIX1* mutation, which was consistent with previous studies showing that *SIX1* variants cause a somewhat alleviated BOR/BO phenotype compared to *EYA1* variants [5,15,17].

### 2.3. Genomic Landscape

Approximately 52% (12 families) of the cohort showed mutation of *EYA1*. Especially, 35% of the cohort showed SNVs in the coding region of *EYA1*, all of which were positioned at the EYA domain (ED); 4% showed SNVs at the canonical splicing region of *EYA1*; and 13% showed SVs, including *EYA1*. SVs of *EYA1* included intragenic deletion (BOR09), complex genomic rearrangement (BOR02), and cryptic inversion (BOR05). WGS and bioinformatics analysis successfully detected two cases (BOR02 and BOR05) characterized by balanced SVs without copy number dosage alterations, which were otherwise challenging to detect with exome sequencing, CNV detection algorithm, and MLPA (Appendix A). These cases exemplify the diagnostic added value of WGS in BOR/BO syndrome. The SV landscape of BOR/BO syndrome, as reviewed in the literature and this study, is depicted in Figure 2. All the SVs reported from cohort studies are shown to affect *EYA1*. Most of the SVs are deletions of *EYA1* (88.9% of SVs), which includes entire or partial deletion of *EYA1*. Complex genomic rearrangement (3.7%), cryptic inversion (3.7%), and Alu element insertion of *EYA1* (3.7%) are reported in a small proportion of SVs landscape. Collectively, these results demonstrate the substantial proportion of SVs underlying BOR/BO syndrome and the potential value of WGS for their detection.

Secondly, 35% (eight families) of the subjects showed *SIX1* variants, all of which were SNVs in the coding region. All of the SNVs of *SIX1,* except for two located in the SIX domain (SD), were positioned in the homeodomain (HD). Lastly, there was a single patient (BOR21) harboring an *ANKRD11* heterozygous variant exhibiting bilateral preauricular fistula, branchial anomalies on the anterior neck, and sensorineural hearing loss on initial examination, which meets three major criteria of BOR/BO syndrome. The patient also had a secundum atrial septal defect (ASD) and fibrolipoma of the filum terminale as additional findings. Exome sequencing identified a pathogenic indel variant (c.2409_2412del) of *ANKRD11,* causing a frameshift change (p.Glu805ArgfsTer57), which in turn leads to nonsense-mediated mRNA decay. *ANKRD11* is known to cause KBG syndrome, characterized by macrodontia of the upper central incisors, distinctive craniofacial findings, short stature, skeletal anomalies, and neurologic involvement, including global developmental delay, seizures, and intellectual disability, which were not seen in our patient [29].

## 3. Discussion

There is growing evidence of the clinical usefulness of WGS in rare diseases. It is well known that WGS can identify variants that are not readily detected by exome sequencing, such as complex rearrangement, tandem repeat expansions, and deep intronic variants [30]. Such diagnostic superiority, as well as increased accessibility of WGS, have begun to replace WES with WGS as a first-line diagnostic test [30,31]. As shown in Figure 1, integrating WGS into the conventional genetic pipeline improved the overall molecular diagnostic yield by detecting complex SVs, which were otherwise challenging to detect with conventional technologies.

In the literature review (Table 2), the overall molecular diagnostic yield for clinically suspicious BOR/BO syndrome patients to date has ranged from 4 to 83%. Most of the studies employed direct sequencing with additional denatured high-performance liquid chromatography (DHPLC) or single-strand conformation polymorphism (SSCP) analysis to detect SNVs of target genes [10,27,32,33]. However, these methods are inadequate for detecting cryptic SVs; hence, additional methods such as Southern blot, MLPA, or quantitative PCR have been adopted to identify SVs. In the systematic literature reviews, the proportion of SVs among total detected variants was approximately 9% (Table 2), and the genomic landscape was illustrated in Figure 2 [10,34]. The rationale behind the frequent occurrence of SVs in BOR/BO syndrome is that non-allelic homologous recombination (NAHR) attributed to human endogenous retrovirus (HERV) elements located near the *EYA1* gene is responsible for a significant portion of SVs [35,36]. Although targeted sequencing or MLPA has demonstrated excellent diagnostic yield for BOR/BO syndrome in the literature [5,37], there are variant types in the same causative genes of BOR/BO syndrome that cannot be detected by conventional targeted approaches. This study aims to improve diagnostic yield in a real-world setting by conducting a deep analysis of the target genes associated with BOR/BO syndrome using WGS. To achieve this, we developed a stepwise genomic pipeline (Figure 1). In this study, WGS successfully identified SVs such as cryptic inversion and complex genomic rearrangement, which were undetectable by conventional sequencing technologies, including exome sequencing, CNV detection algorithm, and MLPA. Our results demonstrate the diagnostic added value of WGS in BOR/BO syndrome, suggesting its potential applicability in the genetic diagnosis of clinically heterogeneous rare diseases.

In line with other genetic disorders, the mutational spectrum of BOR/BO syndrome varies according to genetic ancestry groups. In our cohort, 52% of the subjects exhibited *EYA1* variants, while 35% had *SIX1* variants. This is a slightly higher proportion of *SIX1* variants compared to previously reported values of 3.0 to 4.5% [12,16]. Additionally, neither our cohort nor other Korean cohorts [38] harbored any *SIX5* variants, in contrast to findings in Western populations [13,17].

**Table 2 ijms-25-08149-t002:** Literature review of BOR/BO syndrome cohort studies and this study.

No	PMID	Writer and References	Year	Molecular Diagnostic Yield	Cohort Information	Method	*EYA1*	*SIX1*	*SIX5*	*ANKRD11*	SV Annotation
1	9361030	Sonia Abdelhak et al. [39]	1997	44% (16/36 families)	36 families clinically diagnosed with BOR syndrome	PCR direct exon sequencing Southern blot	16	-	-	-	1 *EYA1* deletion from intron X to exon 16 1 *EYA1* deletion from exon 9 to intron IX 1 *EYA1 Alu* insertion
2	10991693	Sarah Rickard et al. [32]	2000	61% (11/18 families)	18 families with probable BOR syndrome	PCR direct exon sequencing SSCP analysis	11	-	-	-	-
3	15146463	Eugene H. Chang et al. [10]	2004	17% (19/106 families)	106 families with two or more BOR features	SSCP analysis Bidirectional exon sequencing Semi-quantitative fluorescent multiplex PCR	19	-	-	-	2 entire *EYA1* deletions 1 *EYA1* deletion from exon 10 to exon 12
4	16491411	Michiyo Okada et al. [40]	2006	33% (5/15 families)	15 families with BOR syndrome or BOR-related conditions	PCR direct sequencing RT-PCR	5	-	-	-	-
5	17637804	Kirsten Marie Sanggaard et al. [37]	2007	83% (5/6 families)	6 families clinically diagnosed with BOR syndrome	Marker analysis Linkage analysis MLPA PCR direct exon sequencing	4	1	-	-	-
6	17357085	Bethan E. Hoskins et al. [13]	2007	5% (5/95 families)	95 families who met BOR criteria but without *EYA1* or *SIX1* mutations	PCR direct exon sequencing	-	-	5	-	-
7	18330911	Amit Kochhar et al. [33]	2008	4% (10/247 families)	247 families with BOR syndrome	DHPLC Bidirectional exon sequencing PCR direct exon sequencing	-	10	-	-	-
8	18220287	Dana J. Orten et al. [27]	2008	30% (76/248 families)	248 families with at least one of the major BOR criteria	PCR direct exon sequencing DHPLC	76	-	-	-	-
9	19206155	Tracy L. Stockley et al. [41]	2009	82% (14/17 families)	17 families with a clinical suspicion of BOR syndrome	Bidirectional exon sequencing MLPA	14	-	-	-	1 entire *EYA1* deletion 1 *EYA1* deletion of exon 9 1 *EYA1* deletion from exon 9 to exon 10
10	21280147	Krug et al. [17]	2011	46% (45/124 families)	124 families with BOR syndrome	Whole-exome sequencing Multiplex PCR MLPA	42	3	-	-	-
11	22447252	Shin-Hao Wang et al. [42]	2012	16% (2/12 families)	12 families who fulfilled the criteria for BOR syndrome	Direct sequencing of *EYA1*/*SIX1* Quantitative PCR	2	-	-	-	-
12	23840632	Mee Hyun Song et al. [38]	2013	71% (5/7 families)	7 families with hearing loss and one or more typical features of BOR syndrome	PCR direct exon sequencing MLPA	5	-	-	-	1 entire *EYA1* deletion
13	23851940	Patrick D. Brophy et al. [43]	2013	14% (5/32 families)	32 BOR probands negative for coding sequence and splice site mutations in known BOR-causing genes	Array-based CGH Long-range PCR	5	-	-	-	1 *EYA1* deletion from intron 17 to exon 18 and entire 3′ UTR 4 entire *EYA1* deletions
14	28583505	Kyle D. Klingbeil et al. [44]	2017	60% (6/10 families)	10 families clinically diagnosed with BOR syndrome	Whole-exome sequencing Sanger sequencing	6	-	-	-	2 entire *EYA1* deletions
15	29500469	Ai Unzaki et al. [45]	2018	72% (26/36 families)	36 families clinically diagnosed with BOR syndrome	Direct exon sequencing MLPA Array-based CGH NGS	22	1	-	-	1 *EYA1* deletion from exon 10 to exon 18 1 *EYA1* deletion from exon 2 to exon 3 1 *EYA1* deletion from exon 2 to exon 12 1 *EYA1* exon 12 deletion 2 *EYA1* exon 17 deletions
16	31427586	Michie Ideura et al. [26]	2019	32% (19/59 families)	59 families clinically diagnosed with BOR/BO syndrome	NGS Array-based CGH	18	1	-	-	1 entire *EYA1* deletion
	This study	S. H. Cho et al.	2024	91% (21/23 families)	23 families with a clinical suspicion of BOR syndrome	MLPA Whole-exome sequencing Whole-genome sequencing	12	8	-	1	1 *EYA1* complex genomic rearrangement 1 *EYA1* cryptic inversion 1 entire *EYA1* deletion

Abbreviations: SV, structural variation; SSCP analysis, single-stranded conformation polymorphism analysis; DHPLC, denaturing high-performance liquid chromatography; MLPA, multiplex ligation-dependent probe amplification assay; Array-based CGH, array-based comparative genomic hybridization; NGS, next-generation sequencing.

Timely diagnosis and reducing the diagnostic odyssey of rare genetic disorders are crucial for appropriate intervention and improving patient prognosis. In this regard, improved molecular diagnostic yield aided by WGS has significant clinical implications. For example, genetic counseling, including preimplantation genetic diagnosis (PGD) using molecular diagnostics, can be implemented in clinical practice to prevent the transmission of germline variants. Recently, it has been shown that PGD combined with NGS is effective in at-risk populations for preventing the birth of offspring with genetic hearing loss [46]. Moreover, potential therapeutics to edit causative variants could be possible for BOR/BO syndrome. CRISPR-based editing strategies were utilized to rectify complex SVs of the *EYA1* gene [47], extending beyond point mutations. To advance such targeted therapies, including gene therapy, molecular genetic diagnosis is necessary, emphasizing the importance of implementing WGS in real-world practice [48].

In the case of a patient harboring an *ANKRD11* variant (BOR21), KBG syndrome was molecularly diagnosed despite the presence of clinical BOR/BO phenotypes. This case highlights that phenotypic overlap can lead to clinical misdiagnosis of syndromic diseases. Not only KBG syndrome but also Townes–Brocks syndrome caused by *SALL1* variants and branchio-oculo-facial syndrome (BOFS) caused by *TFAP2A* variants show pheno-typic overlap with BOR/BO syndrome [49,50,51,52]. Therefore, it is necessary to differentiate these conditions from BOR/BO syndrome, highlighting the important role of molecular diagnostics. Unlike targeted panel sequencing, WGS can capture the entire genome, enabling us to screen not only suspected genes but also other genes. To our knowledge, the functional pathogenicity of *ANKRD11* variants is not intercorrelated with the EYA1-SIX1-DNA theory underlying the pathogenesis of BOR/BO syndrome. It is possible that molecular pathways beyond the genomic sequence, such as epigenetic mechanisms, might interplay to explain the phenotypic overlap of KBG syndrome and BOR/BO syndrome, similar to the overlap observed between CHARGE and Kabuki syndromes [53].

## 4. Materials and Methods

### 4.1. Participants and Clinical Assessment

All procedures in this study were approved by the Institutional Review Boards of Seoul National University Hospital (IRB-H-2202-045-1298). A total of 41 patients from 23 unrelated families who were clinically suspected of having BOR/BO syndrome were enrolled. All participants were attending the Hereditary Hearing Loss Clinic within the Otorhinolaryngology division at the Center for Rare Diseases, Seoul National University Hospital, South Korea. The demographic and clinical phenotypes of the subjects were retrieved from electronic medical records, including underlying disease history, physical examinations, radiological imaging, otoendoscope findings, and audiological evaluations.

### 4.2. Conventional Genetic Pipeline

Genomic DNA from the subjects was extracted from peripheral blood samples using the Chemagic 360 instrument (Perkin Elmer, Baesweiler, Germany). Whole-exome sequencing was conducted using SureSelectXT Human All Exon V5 (Agilent Technologies, Santa Clara, CA, USA). Sequence reads were aligned to the human reference genome (GRCh38) and analyzed with Genome Analysis Toolkit (GATK) [54] to detect single-nucleotide variations (SNVs) and indels. As previously described [15,55,56,57,58], bioinformatics analysis and stringent filtering based on following criteria were performed: (i) Selecting non-synonymous variants with quality scores > 30 and read depths > 10. (ii) Filtering variants with minor allele frequencies (MAFs) ≤ 0.001 based on population database, including gnomAD (v.4.1.0; https://gnomad.broadinstitute.org/; accessed on 1 May 2024) and a reference database of genetic variations in the Korean population (KOVA2) (v.2; https://www.kobic.re.kr/kova/; accessed on 1 May 2024). (iii) Assessing variant pathogenicity using in silico tools, including Combined Annotation Dependent Depletion (CADD) (v.1.7; https://cadd.gs.washington.edu/; accessed on 1 May 2024) and Rare Exome Variant Ensemble Learner (REVEL) (https://sites.google.com/site/revelgenomics/; accessed on 1 May 2024). (iv) Evaluating inheritance patterns and audiological/clinical phenotypes of variants and screening ClinVar and HGMD databases to check whether candidate variants had been previously identified in other patients. (v) Confirming candidate variants through Sanger sequencing and conducting segregation study via trio-sequencing using proband and parental DNA samples. Collectively, the pathogenicity of candidate variants was evaluated according to the ACMG-AMP guidelines for SNHL [59]. In cases of inconclusive whole-exome sequencing results, multiplex ligation-dependent probe amplification (MLPA) was conducted. In parallel, copy number variations (CNVs) were detected using two exome sequencing-based algorithms: CNVkit [60] and Copy Number Inference from Exome Reads (CoNIFER) [61]. We also visually inspected copy number changes in candidate regions identified by these detection algorithms using the Integrative Genomics Viewer (IGV).

### 4.3. Whole-Genome Sequencing and Bioinformatic Analysis

DNA libraries for whole-genome sequencing (WGS) were prepared using the TruSeq DNA PCR-Free kit (Illumina, San Diego, CA, USA) with 1 µg of genomic DNA. Sequencing was performed on the NovaSeq 6000 platform (Illumina) to generate 151 bp paired-end reads. The raw sequencing data and subsequent analysis were managed using RareVision™ (Genome Insight, Inc., Daejeon, Republic of Korea). Reads were aligned to the GRCh38 reference genome using the BWA-MEM algorithm, followed by removal of duplicate reads with SAMBLASTER [62]. HaplotypeCaller [54] and Strelka2 [63] were used to detect base substitutions and short indels, and Delly [64] was employed to identify genomic rearrangements. The breakpoints of selected genomic rearrangements were visually inspected and confirmed using the Integrative Genomics Viewer (IGV).

## 5. Conclusions

We investigated the genomic landscape of 23 unrelated Korean families with typical or atypical BOR/BO syndrome. By integrating WGS into our diagnostic pipeline, we detected an array of structural variations, including complex genomic rearrangements and cryptic inversions, ultimately increasing the diagnostic yield to 91%. This comprehensive analytical framework has significantly enhanced the molecular diagnostic yield of BOR/BO syndrome compared to previous studies. Our findings suggest the clinical utility of WGS in diagnosing rare diseases, including those with BOR/BO syndrome.

## Figures and Tables

**Figure 1 ijms-25-08149-f001:**
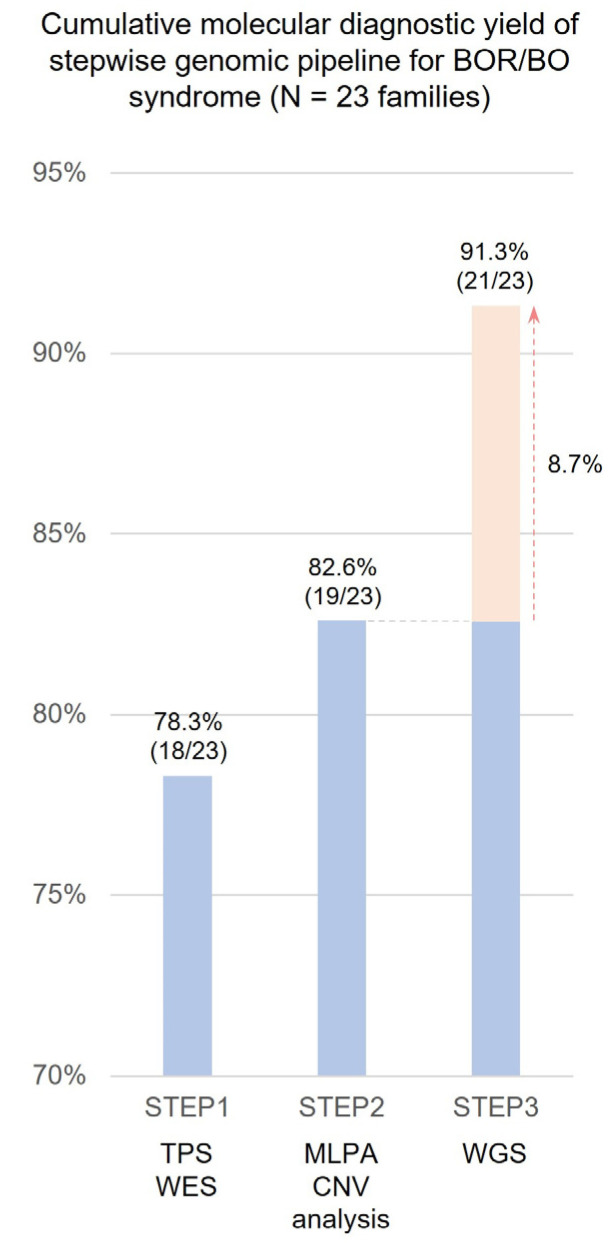
Cumulative molecular diagnostic yield of stepwise genomic pipeline. Targeted panel sequencing and exome sequencing were initially utilized to identify causative variants (Step 1). For patients who remained undiagnosed after targeted panel sequencing and exome sequencing, exome sequencing-based CNV screening coupled with MLPA (Step 2) was applied. Finally, WGS (Step 3) was applied to patients who remained genetically undiagnosed after Step 2.

**Figure 2 ijms-25-08149-f002:**
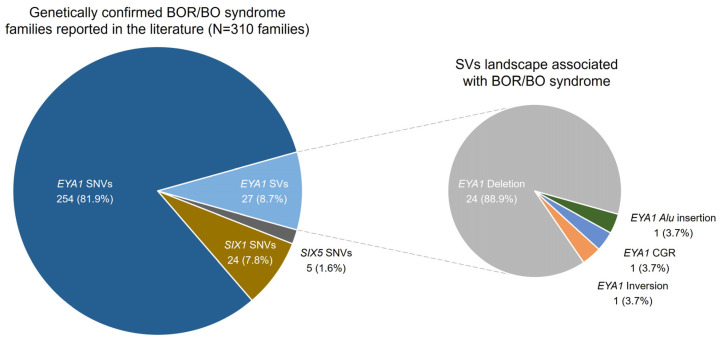
SVs landscape of BOR/BO syndrome from literature review of BOR/BO syndrome cohort studies and this study. SVs accounted for 8.7% of all mutations reported in the cohort studies. Most of the SVs are deletions of *EYA1* (89% of SVs), while complex genomic rearrangement (4% of SVs) and inversion of *EYA1* (4% of SVs) are reported in a small proportion of total SVs.

**Table 1 ijms-25-08149-t001:** Cohort description, genotypes, and clinical phenotypes in BOR/BO syndrome.

Family	Sex/Age	Diagnostic Approach *	Gene	Variant [NM/NP No.]	Zygosity/ Inheritance	ACMG Classification ^#^	Affected Domain	Branchial Anomalies	Preauricular Pits	Hearing Loss	Renal Anomalies	EAC Anomalies	Middle Ear Anomalies	Inner Ear Anomalies	Typical/ Atypical
BOR01	F/15	WES	*EYA1*	c.1319G>A;p.Arg440Gln [NM_000503.6/NP_000494.2]	Het/ de novo	Pathogenic	ED	O	O	MHL	O	X	O	O	Typical
BOR02	F/22	WGS	*EYA1*	Complex genomic rearrangement g.[71211857_712282326inv;712111857_71215145del]	Het/AD	Pathogenic	N/D	O	O	MHL	X	X	O	O	Typical
	F/56	WGS	*EYA1*	Complex genomic rearrangement g.[71211857_712282326inv;712111857_71215145del]	Het/AD	Pathogenic	N/D	X	O	MHL	X	O	O	X	Atypical
	M/32	WGS	*EYA1*	Complex genomic rearrangement g.[71211857_712282326inv;712111857_71215145del]	Het/AD	Pathogenic	N/D	O	O	SNHL	O	X	X	O	Typical
	F/29	WGS	*EYA1*	Complex genomic rearrangement g.[71211857_712282326inv;712111857_71215145del]	Het/AD	Pathogenic	N/D	O	O	SNHL	X	X	X	O	Typical
BOR03	M/31	WES	*EYA1*	c.1623_1626dup:p.Gln543AsnfsTer90 [NM_000503.6/NP_000494.2]	Het/ de novo	Pathogenic	ED	O	O	SNHL	X	X	X	O	Typical
BOR04	F/0	WES	*EYA1*	c.1598-2A>C:p.? [NM_000503.6/NP_000494.2]	Het/AD	Pathogenic	N/D	O	O	MHL	X	O	O	X	Typical
	F/30	WES	*EYA1*	c.1598-2A>C:p.? [NM_000503.6/NP_000494.2]	Het/AD	Pathogenic	N/D	O	O	MHL	X	X	O	X	Typical
BOR05	F/4	WGS	*EYA1*	Cryptic inversion c.49-7047[NC_000008.11:g.71448124]inv	Het/AD	Pathogenic	N/D	O	O	SNHL	X	O	X	O	Typical
	F/33	WGS	*EYA1*	Cryptic inversion c.49-7047[NC_000008.11:g.71448124]inv	Het/AD	Pathogenic	N/D	O	X	MHL	X	X	O	X	Typical
BOR06	F/8	WES	*EYA1*	c.1081C>T:p.Arg361Ter [NM_000503.6/NP_000494.2]	Het/AD	Pathogenic	ED	O	O	MHL	X	O	O	O	Typical
	M/8	WES	*EYA1*	c.1081C>T:p.Arg361Ter [NM_000503.6/NP_000494.2]	Het/AD	Pathogenic	ED	O	O	MHL	X	X	O	O	Typical
BOR07	M/17	WES	*EYA1*	c.1220G>A:p.Arg407Gln [NM_000503.6/NP_000494.2]	Het / de novo	Pathogenic	ED	O	O	MHL	O	O	O	O	Typical
BOR08	M/6	WES	*EYA1*	c.1276G>A:p.Gly426Ser [NM_000503.6/NP_000494.2]	Het/AD	Likely Pathogenic	ED	O	O	MHL	O	O	O	X	Typical
	M/9	WES	*EYA1*	c.1276G>A:p.Gly426Ser [NM_000503.6/NP_000494.2]	Het/AD	Likely Pathogenic	ED	O	O	X	X	X	X	X	Atypical
BOR09	F/12	MLPA	*EYA1*	Deletion	Het/AD	Pathogenic	N/D	O	O	MHL	X	O	O	O	Typical
BOR10	M/7	WES	*EYA1*	c.1081C>T:p.Arg361Ter [NM_000503.6/NP_000494.2]	Het/AD	Pathogenic	ED	X	O	MHL	O	O	O	O	Typical
BOR11	F/7	WES	*EYA1*	c.1715G>A:p.Trp572Ter [NM_000503.6/NP_000494.2]	Het/AD	Likely Pathogenic	ED	O	O	MHL	X	X	O	O	Typical
	F/12	WES	*EYA1*	c.1715G>A:p.Trp572Ter [NM_000503.6/NP_000494.2]	Het/AD	Likely Pathogenic	ED	O	O	MHL	X	X	O	X	Typical
BOR12	M/31	WES	*EYA1*	c.802C>T:p.Gln268Ter [NM_000503.6/NP_000494.2]	Het/AD	Likely Pathogenic	ED	O	O	MHL	X	X	O	X	Typical
	M/28	WES	*EYA1*	c.802C>T:p.Gln268Ter [NM_000503.6/NP_000494.2]	Het/AD	Likely Pathogenic	ED	O	O	MHL	X	X	O	X	Typical
	M/64	WES	*EYA1*	c.802C>T:p.Gln268Ter [NM_000503.6/NP_000494.2]	Het/AD	Likely Pathogenic	ED	O	O	SNHL	X	X	X	O	Typical
	F/56	WES	*EYA1*	c.802C>T:p.Gln268Ter [NM_000503.6/NP_000494.2]	Het/AD	Likely Pathogenic	ED	O	O	MHL	X	X	O	O	Typical
	F/49	WES	*EYA1*	c.802C>T:p.Gln268Ter [NM_000503.6/NP_000494.2]	Het/AD	Likely Pathogenic	ED	O	O	MHL	X	X	O	X	Typical
	F/20	WES	*EYA1*	c.802C>T:p.Gln268Ter [NM_000503.6/NP_000494.2]	Het/AD	Likely Pathogenic	ED	O	O	MHL	X	X	O	X	Typical
BOR13	F/31	WES	*SIX1*	c.501G>C:p.Gln167His [NM_005982.4/NP_005973.1]	Het/AD	Likely Pathogenic	HD	O	X	SNHL	X	X	X	X	Atypical
	F/60	WES	*SIX1*	c.501G>C:p.Gln167His [NM_005982.4/NP_005973.1]	Het/AD	Likely Pathogenic	HD	X	X	SNHL	X	X	X	X	Atypical
	F/30	WES	*SIX1*	c.501G>C:p.Gln167His [NM_005982.4/NP_005973.1]	Het/AD	Likely Pathogenic	HD	X	X	SNHL	X	X	X	X	Atypical
BOR14	F/10	WES	*SIX1*	c.386_391del:p.Tyr129_Cys130del [NM_005982.4/NP_005973.1]	Het/AD	Pathogenic	HD	X	O	SNHL	X	X	X	X	Atypical
	F/40	WES	*SIX1*	c.386_391del:p.Tyr129_Cys130del [NM_005982.4/NP_005973.1]	Het/AD	Pathogenic	HD	X	O	SNHL	X	X	X	X	Atypical
BOR15	F/11	WES	*SIX1*	c.397_399del:p.Glu133del [NM_005982.4/NP_005973.1]	Het/ de novo	Likely Pathogenic	HD	X	O	SNHL	X	X	X	O	Atypical
BOR16	M/76	WES	*SIX1*	c.21del:p.Phe7LeufsTer82 [NM_005982.4/NP_005973.1]	Het/AD	Pathogenic	SD	X	O	SNHL	X	X	X	X	Atypical
BOR17	M/10	WES	*SIX1*	c.386A>C:p.Tyr129Ser [NM_005982.4/NP_005973.1]	Het/AD	Likely Pathogenic	HD	X	X	SNHL	X	X	X	X	Atypical
	M/44	WES	*SIX1*	c.386A>C:p.Tyr129Ser [NM_005982.4/NP_005973.1]	Het/AD	Likely Pathogenic	HD	X	X	MHL	X	X	O	X	Atypical
BOR18	M/22	WES	*SIX1*	c.176A>C:p.His59Pro [NM_005982.4/NP_005973.1]	Het/AD	Likely Pathogenic	SD	X	O	SNHL	X	X	X	X	Atypical
BOR19	M/16	WES	*SIX1*	c.513G>T:p.Trp171Cys [NM_005982.4/NP_005973.1]	Het/ de novo	Pathogenic	HD	O	O	CHL	X	X	X	X	Typical
BOR20	M/1	WES	*SIX1*	c.376_378del:p.Glu126del [NM_005982.4/NP_005973.1]	Het/AD	Likely Pathogenic	HD	X	O	SNHL	X	X	X	X	Atypical
	M/1	WES	*SIX1*	c.376_378del:p.Glu126del [NM_005982.4/NP_005973.1]	Het/AD	Likely Pathogenic	HD	X	X	SNHL	X	X	X	X	Atypical
BOR21	F/1	WES	*ANKRD11*	c.2409_2412del:p.Glu805ArgfsTer57 [NM_013275.6/NP_037407.4]	Het/AD	Pathogenic	Linker region	O	O	SNHL	X	X	X	X	Typical
BOR22	M/13	WGS	Negative	-	-	-	-	O	O	MHL	O	X	O	O	Typical
BOR23	F/51	WES	Negative	-	-	-	-	X	O	MHL	X	X	O	X	N/D

Abbreviations: Het, heterozygote; AD, autosomal dominant; de novo, de novo confirmed; SNHL, sensorineural hearing loss; MHL, mixed hearing loss; ED, EYA domain; HD, homeodomain; SD, SIX domain; N/D, not determined; WES, whole-exome sequencing; WGS, whole-genome sequencing. * Sequencing technology that identifies the causative variants. # ACMG/AMP 2015 guideline (http://wintervar.wglab.org/; accessed on 1 May 2024).

## Data Availability

The original contributions presented in the study are included in the article/Appendix A, further inquiries can be directed to the corresponding author.

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
