# Peer review of "Genomic Landscape of Branchio-Oto-Renal Syndrome through Whole-Genome Sequencing: A Single Rare Disease Center Experience in South Korea"

_ijms, 2024, doi:10.3390/ijms25158149_

Round 1

Reviewer 1 Report

Comments and Suggestions for Authors

Authors present a screening study in 23 families to find out unerdlying muations of BOR / BO syndrome.

They  did  stepwise diagnostics and found mainly mutations / CNVs in genes EYA1 and SIX1

As work is done in Korean instead of Europeans the paper is of importance

major comments

- make a celar statement in discussion if mutation spectrum found in Korans is different than in Europeans

- explain why you seem to suggest to use your approach of stepwise diagnostics, instead of screening for mutations found in this study in Korean/ Asian BOR/BO families by MLPA or Sanger-sequencing. 

- donot include the pseudo-argument that WGS/WES is sooo cheap - technically it may be relatively cheap - but you have always to consider and take into account manpower for interpretation of data, which is only then negectable if results are glass clear - however, they are not in majority of cases tested still.  Also you should include into your cost calculations the costs for buying the mashine and also for yearly service of 50.000 Euro and more, being also needed to fullfill needs of quality management. 

- provide a Conclusion part

minor comments

- patients seem to have lab internal identifiers / numbers - this needs to be changes to anonymous numbers within paper and suppl. file

- gene names need to be written in italics also in result part

Author Response

Major comments

Comments 1) make a clear statement in discussion if mutation spectrum found in Korans is different than in Europeans

[Response] We appreciate your feedback. We have updated the manuscript to include information about the difference in mutational spectrum of BOR/BO syndrome between Koreans and Europeans. Please refer to lines 203-208:

“In line with other genetic disorders, the mutational spectrum of BOR/BO syndrome varies according to genetic ancestry groups. In our cohort, 52% of the subjects exhibited EYA1 variants, while 35% had SIX1 variants. This is a slightly higher proportion of SIX1 variants compared to previously reported values of 3.0 to 4.5% [12,15]. Additionally, neither our cohort nor other Korean cohorts [37] harbored any SIX5 variants, in contrast to findings in Western populations [13,17].”

Comments 2) explain why you seem to suggest to use your approach of stepwise diagnostics, instead of screening for mutations found in this study in Korean/Asian BOR/BO families by MLPA or Sanger-sequencing: Need to be included

[Response] We are grateful for the reviewer’s comments. To make the meaning of the text clear, we have modified some parts of the text. Please refer to lines 191-202:

“Although targeted sequencing or MLPA have demonstrated excellent diagnostic yield for BOR/BO syndrome in the literature [5,36], there are variant types in the same causative genes of BOR/BO syndrome that cannot be detected by conventional targeted approaches. This study aims to improve diagnostic yield in a real-world setting by conducting a deep analysis of the target genes associated with BOR/BO syndrome using WGS. To achieve this, we developed a stepwise genomic pipeline. In this study, WGS successfully identified SVs such as cryptic inversion and complex genomic rearrangement, which were undetectable by conventional sequencing technologies including exome sequencing, CNV detection algorithm, and MLPA. Our results demonstrate the diagnostic added value of WGS in BOR/BO syndrome, suggesting its potential applicability in the genetic diagnosis of clinically heterogeneous rare diseases.”

Comments 3) don’t include the pseudo-argument that WGS/WES is so cheap - technically it may be relatively cheap - but you have always to consider and take into account manpower for interpretation of data, which is only then neglectable if results are glass clear - however, they are not in majority of cases tested still. Also, you should include into your cost calculations the costs for buying the machine and also for yearly service of 50.000 Euro and more, being also needed to fulfill needs of quality management

[Response] Thank you for constructive comments. All in all, we agree with the reviewer’s comments. Although recent studies have shown the utility of WGS for the genetic diagnosis of several disorders, even as a first-line test, WGS is not yet widely applied for real-world clinical use due to several limitations, including the difficulties of bioinformatic analysis and accurate clinical interpretation. Since this study does not aim to evaluate the cost-effectiveness of WGS, all related sentences will be removed.

Comments 4) provide a Conclusion part

[Response] Thank you for your comments. We included conclusion part in our manuscript. Please refer to lines 289-295:

“We investigated the genomic landscape of 23 unrelated Korean families with typical or atypical BOR/BO syndrome. By integrating WGS into our diagnostic pipeline, we detected an array of structural variations, including complex genomic rearrangements and cryptic inversions, ultimately increasing the diagnostic yield to 91%. This comprehensive analytical framework has significantly enhanced the molecular diagnostic yield of BOR/BO syndrome compared to previous studies. Our findings suggest the clinical utility of WGS in diagnosing rare diseases, including those with BOR/BO syndrome.”

Minor comments

Comments 5) patients seem to have lab internal identifiers / numbers - this needs to be changes to anonymous numbers within paper and suppl. File

[Response] Thank you for pointing out the details. We have anonymized the lab internal identifiers and numbers in our text. We changed all instances of “SNUH family No.” to “BOR family No.”

Comments 6) gene names need to be written in italics also in result part

[Response] Thank you for the detailed comments. We have italicized the gene names in our script.

Reviewer 2 Report

Comments and Suggestions for Authors

In the manuscript entitled "Genomic Landscape of Brancio-Oto-Renal Syndrome through Whole-Genome Sequencing: A Single Rare Disease Center Experience in South Korea" Sung Ho Cho et al. outlined the significance of Whole Genome Sequencing contribution in the genetic diagnosis of rare diseases with complex genomic rearrangements in a satisfactory manner.

However, the authors are asked to elaborate on the following comments:

1) It will be useful to mention the known function of the genes (i.e. EYA1, SIX1) associated with BOR/BO pathogenicity.

2) What are the most frequent types of pathogenic mutations (e.g. SNVs, indels) associated with this condition?

3) Modify the margins of Table 1.

4) Were there any Variants of Unknown Significance identified by WGS that could be potential candidates for BOR/BO pathogenicity?

5) What is the sequencing depth that was used for WES and WGS?

6) Please add versions to the databases and bioinformatics tools used for sequencing variant annotation and curation.

Author Response

Comments 1) It will be useful to mention the known function of the genes (i.e. EYA1, SIX1) associated with BOR/BO pathogenicity

[Response] Thank you for your feedback, which makes our script clearer. We have added the information about the known function of BOR/BO syndrome causing genes. Please refer to lines 51-55:

“It is well known that EYA1 binds to SIX1 and SIX5 to form a bipartite transcription factor [14]. Especially, SIX1 protein binds to the Eya domain of EYA1 using its Six domain and concurrently binds to DNA element with its DNA binding homeodomain to form EYA1-SIX1-DNA complex [15]. In turn, EYA1-SIX1-DNA complex regulates organogenesis including branchial arch, otic and renal systems [10,15].”

Comments 2) What are the most frequent types of pathogenic mutations (e.g. SNVs, indels) associated with this condition?

[Response] Thank you for your comments. As shown in Figure 2, most frequent types of pathogenic mutations associated with BOR/BO syndrome is SNVs of EYA1 gene.

Comments 3) Modify the margins of Table 1 : OK

[Response] Thank you for your detailed comment. We have modified the font size of margins of Table 1 to enhance readability of our text.

Comments 4) Were there any Variants of Unknown Significance identified by WGS that could be potential candidates for BOR/BO pathogenicity?

[Response] We appreciate for your feedback.

We identified causative SVs for BOR/BO syndrome in 2 out of 4 undiagnosed patients who underwent WGS. Still, for the remaining 2 undiagnosed patients after WGS, neither VUS in BOR/BO target genes nor in any genes within the BOR/BO loci were detected. As you pinpointed, there may be potential candidate genes outside the BOR/BO loci. However, at this stage, VUS (at least meet PM2) were ruled out based on the OMIM database as they did not match the phenotypes. If additional undiagnosed BOR/BO patients are accumulated after WGS, we may be able to identify new potential candidate genes through overlapping sequencing strategies.

Comments 5) What is the sequencing depth that was used for WES and WGS?

[Response] Thank you for your comment. The average depth of coverage for WES and WGS was 100x and 30x, respectively.

Comments 6) Please add versions to the databases and bioinformatics tools used for sequencing variant annotation and curation

[Response] We are grateful for the reviewer’s comments. We have added the versions of the database and bioinformatics tools used in our study.